# Role of Human Papillomavirus Infection in Head and Neck Cancer in Italy: The HPV-AHEAD Study

**DOI:** 10.3390/cancers12123567

**Published:** 2020-11-29

**Authors:** Marta Tagliabue, Marisa Mena, Fausto Maffini, Tarik Gheit, Beatriz Quirós Blasco, Dana Holzinger, Sara Tous, Daniele Scelsi, Debora Riva, Enrica Grosso, Francesco Chu, Eric Lucas, Ruediger Ridder, Susanne Rrehm, Johannes Paul Bogers, Daniela Lepanto, Belén Lloveras Rubio, Rekha Vijay Kumar, Nitin Gangane, Omar Clavero, Michael Pawlita, Devasena Anantharaman, Madhavan Radhakrishna Pillai, Paul Brennan, Rengaswamy Sankaranarayanan, Marc Arbyn, Francesca Lombardi, Miren Taberna, Sara Gandini, Fausto Chiesa, Mohssen Ansarin, Laia Alemany, Massimo Tommasino, Susanna Chiocca

**Affiliations:** 1Division of Otolaryngology and Head and Neck Surgery, IEO, European Institute of Oncology IRCCS, 20141 Milan, Italy; marta.tagliabue@ieo.it (M.T.); daniele.scelsi@gmail.com (D.S.); deborariva88@gmail.com (D.R.); enrica.grosso@ieo.it (E.G.); francesco.chu@ieo.it (F.C.); faustochiesa@gmail.com (F.C.); mohssen.ansarin@ieo.it (M.A.); 2Cancer Epidemiology Research Program, Catalan Institute of Oncology-Bellvitge Biomedical Research Institute (ICO-IDIBELL), L’Hospitalet de Llobregat, 08908 Barcenola, Spain; mmena.iconcologia@gmail.com (M.M.); bquiros@iconcologia.net (B.Q.B.); stous@iconcologia.net (S.T.); oclavero@iconcologia.net (O.C.); lalemany@iconcologia.net (L.A.); 3Centro de Investigación Biomédica en Red: Epidemiología y Salud Pública (CIBERESP), Instituto de Salud Carlos III, 28029 Madrid, Spain; 4Division of Pathology, IEO, European Institute of Oncology IRCCS, 20141 Milan, Italy; fausto.maffini@ieo.it (F.M.); daniela.lepanto@ieo.it (D.L.); 5Infections and Cancer Biology Group, International Agency for Research on Cancer (IARC), 69372 Lyon, France; ghett@iarc.fr (T.G.); LucasE@iarc.fr (E.L.); 6Deutsches Krebsforschungszentrum (DKFZ), 69120 Heidelberg, Germany; dana.holzinger@gmx.de (D.H.); m.pawlita@dkfz.de (M.P.); 7Roche mtm laboratories, 69117 Mannheim, Germany; ruediger.ridder@roche.com (R.R.); susanne.rehm@roche.com (S.R.); 8Ventana Medical Systems Inc./Roche Tissue Diagnostics, Tucson, AZ 85755, USA; 9Laboratory for Cell Biology and Histology, University of Antwerp, 2610 Antwerp, Belgium; john-paul.bogers@uantwerpen.be; 10Hospital del Mar, 08003 Barcelona, Spain; BLloveras@parcdesalutmar.cat; 11Kidwai Memorial Institute of Oncology, Bangalore, Karnataka 560029, India; rekha_v_kumar@yahoo.co.in; 12Mahatma Gandhi Institute of Medical Sciences, Sevagram, Wardha, Maharashtra State 442102, India; nitingangane@gmail.com; 13Rajiv Gandhi Centre for Biotechnology, Poojappura, Thiruvananthapuram, Kerala 695012, India; devasenaa@gmail.com (D.A.); mrpillai@rgcb.res.in (M.R.P.); 14Section of Genetics, International Agency for Research on Cancer (IARC), 69372 Lyon, France; BrennanP@iarc.fr; 15Research Triangle Institute (RTI) International India, New Delhi 110001, India; sankardr@hotmail.com; 16Unit of Cancer Epidemiology/Belgian Cancer Centre, Sciensano, 1050 Brussels, Belgium; marc.arbyn@sciensano.be; 17Data Management, IEO, European Institute of Oncology IRCCS, 20141 Milan, Italy; francesca.lombardi@ieo.it; 18Medical Oncology Department, Catalan Institute of Oncology (ICO), ONCOBELL, IDIBELL, L’Hospitalet de Llobregat, 08035 Barcelona, Spain; mtaberna@iconcologia.net; 19Department of Experimental Oncology, IEO, European Institute of Oncology IRCCS, 20141 Milan, Italy; sara.gandini@ieo.it

**Keywords:** head and neck cancer, human papillomavirus, oropharyngeal cancer, virus-related cancers, human papillomavirus diagnosis

## Abstract

**Simple Summary:**

This is the largest and most comprehensive assessment of the role of human papillomavirus (HPV) in head and neck cancer (HNC) in Italy, which is a region currently considered bearing a low burden of HPV-driven HNC. p16^INK4a^, HPV-DNA, and HPV RNA biomarkers were used to assess the HPV status in head and neck cancer in a retrospective cohort of approximately 700 patients. In our study, HPV prevalence in oropharyngeal cancers was much higher than in oral and laryngeal cancers, and HPV positivity conferred better prognosis only in oropharyngeal cancers. Importantly, we have observed an increase of the prevalence of HPV positivity in oropharyngeal cancers in the most recent calendar periods, suggesting that this disease is increasing in Italy, as has happened before in other developed regions.

**Abstract:**

Literature on the role of human papillomavirus (HPV) in head and neck cancer (HNC) in Italy is limited, especially for non-oropharyngeal tumours. Within the context of the HPV-AHEAD study, we aimed to assess the prognostic value of different tests or test algorithms judging HPV carcinogenicity in HNC and factors related to HPV positivity at the European Institute of Oncology. We conducted a retrospective cohort study (2000–2010) on a total of 696 primary HNC patients. Formalin-fixed, paraffin-embedded cancer tissues were studied. All HPV-DNA-positive and a random sample of HPV-DNA-negative cases were subjected to HPV-E6*I mRNA detection and p16^INK4a^ staining. Multivariate models were used to assess for factors associated with HPV positivity and proportional hazards for survival and recurrence. The percentage of HPV-driven cases (considering HPV-E6*I mRNA positivity) was 1.8, 2.2, and 40.4% for oral cavity (OC), laryngeal (LC), and oropharyngeal (OPC) cases, respectively. The estimates were similar for HPV-DNA/p16^INK4a^ double positivity. Being a non-smoker or former smoker or diagnosed at more recent calendar periods were associated with HPV-E6*I mRNA positivity only in OPC. Being younger was associated with HPV-E6*I mRNA positivity in LC. HPV-driven OPC, but not HPV-driven OC and LC, showed better 5 year overall and disease-free survival. Our data show that HPV prevalence in OPC was much higher than in OC and LC and observed to increase in most recent years. Moreover, HPV positivity conferred better prognosis only in OPC. Novel insights on the role of HPV in HNC in Italy are provided, with possible implications in the clinical management of these patients.

## 1. Introduction

Head and neck carcinoma (HNC) is a heterogeneous group of tumours located at the nasopharynx, oropharynx, hypopharynx, larynx, and oral cavity. Over 90% of HNC are squamous cell carcinomas (HNSCC) and are caused mainly by environmental factors such as smoking, alcohol consumption, human papillomavirus (HPV), and Epstein Barr (EBV) infections. According to the Surveillance, Epidemiology, and End Results (SEER), approximately 4% of all worldwide cancers are in the head and neck region, with more than 430,000 cases per year [1].

Globocan future estimations of HNC in Italy are projected to increase: specifically, a boost of more than 10% of new cases is estimated in all head and neck (HN) areas for both sexes, at all ages, per 100,000 people [2], within the next decade. Compared to 2018, in 2030 the incidence is predicted to increase by 10% for oropharyngeal cancers (OPC) and by 12, 14, and 16% for oral cavity (OC), larynx (LC), and hypopharynx (HPC) cancers, respectively, regardless of HPV status [2]. In addition, a cancer registry-based study assessing the incidence and survival patterns of HNC diagnosed in Italy between 1988 and 2012 found increasing incidence rates of OPC, presumably attributed to HPV infection [3]. Likewise, increasing trends of HPV-driven OPC have been observed for the last two decades in other parts of the world [4].

The HPV distribution in HNSCC largely differs by anatomical site: while the prevalence of HPV-driven OPC ranges between 30–40%, much lower are the estimated rates for the other areas, specifically 2.1–4.4% for OC and 2.7–3.5% for LC and HPC [5,6,7,8,9].

Italian data on HPV status in HNSCC were reported from three other studies with HPV-DNA/HPV-E6*I mRNA double positivity with variable ranges such as 37.9% in OPC (1992–2015) [6], 6% in OC, 20% in OPC, and 1% in LC (2003–2012) [10], and 32.3% in more recent years (2000–2018) for OPC [11]. Indeed, in Italy, HPV infection plays a role in HNSCC similarly to the rest of the Western world, not only for its well documented positive prognostic value but also for its mediation in carcinogenesis [12,13]. However, currently, it is crucial to fully understand how the presence of HPV interacts with other risk factors such as smoking and alcohol in HNSCC in this region [14,15]. From the etiological point of view and in terms of tumour progression and prognosis, the role of these known risk factors is still debated, and those are considered as either “HPV competitors” or as positively interacting with HPV [16,17,18].

The distinction between HPV-driven and non-HPV-driven OPC is underscored also in the 8th edition of the TNM, where the HPV status (as defined by p16 ^INK4a^ staining) has been considered for the stage classification of the tumour [19]. It is thus critical to select and use robustly sensitive and specific HPV diagnostic assays in order to determine whether the tumour is truly an HPV-driven OPC. The mere detection of HPV-DNA could reflect a transient or non-related infection rather than a genuine HPV-driven oncogenic process [20,21,22]. Several markers have been described and are used for HPV detection in HNC, such as E6/E7 HPV mRNA RT-PCR, HPV-DNA/RNA in situ hybridisation, and p16^INK4a^ immunohistochemistry (IHC) [23]. The identification of viral E6/E7 mRNA [22] is widely accepted as the present gold-standard test to elucidate the oncogenic role of HPV but is difficult to employ in everyday clinical practice. Cellular p16^INK4a^ high expression detected by IHC is the most widely implemented technique in clinical settings for HPV-driven OPC diagnosis [24]. However, a significant fraction of p16^INK4a^-positive OPCs are HPV-DNA-negative with no prognostic advantage with respect to HPV-DNA/p16^INK4a^ double-negative tumours, as they might not be related to HPV [25]. The combination of HPV-DNA detection and p16^INK4a^ IHC is starting to be recommended to diagnose HPV-related OPCs [26]. Outside the oropharynx, p16^INK4a^ IHC is not recommended for the diagnosis of HPV association; however, there is limited information about the accuracy and prognostic value of dual HPV-DNA and p16^INK4a^ testing in non-oropharyngeal HNSCC.

Moreover, country-specific estimates of HPV-attributable fractions in OPC and non-oropharyngeal HNC are warranted in order to evaluate the possible protective effect of HPV vaccination.

In this study, we assessed the prognostic value of HPV positivity (as defined by HPV-E6*I mRNA positivity) in a sample of OPC and non-OPC Italian patients and compared the results with those from HPV-DNA positivity and HPV-DNA/p16^INK4a^ double positivity.

We also studied factors related to HPV positivity (as defined by different HPV-relatedness definitions) and the overall proportion and type distribution of HPV-positive at different anatomical sites, as well as the trend of the proportion of HPV-positive HNSCC in more recent years, in Italy. Finally, we highlight the differences between HPV-positive versus negative cancers at three different anatomical regions in terms of prognosis and survival in an Italian setting.

## 2. Results

### 2.1. HPV Type Distribution in HPV-Driven HN Sites According to Different Combination of Biomarkers

Appendix A shows the workflow of the HNSCC cases, samples collected, processed, tested, and finally included in the study. 

A total of 1594 cases consecutively diagnosed with a primary HNSCC at European Institute of Oncology, IRCCS (IEO) in 2000–2010 were identified, of which 835 (52%) had unavailable formalin-fixed, paraffin-embedded (FFPE) tissue blocks at diagnosis. Some differences were observed between cases with and without available FFPE tissue blocks: specimens from younger patients, cases diagnosed with stage IV a-b (7th TNM edition) or located at sites distal to the oropharynx were over-represented among OC cases with available FFPE tissue blocks. Cases diagnosed with stage III (7th TNM edition) were over-represented among OPC cases with available FFPE tissue blocks, whereas OPC cases located at the base of the tongue were under-represented. Smokers and drinkers, as well as cases diagnosed at earlier periods and located at anatomical subsites proximal to the oropharynx, were over-represented among LC cases with available FFPE tissue blocks. A total of 696 primary cases had a valid HPV result and 675 were finally included in the analyses: 165 OC, 109 OPC, and 401 LC cases (21 pharyngeal-hypopharyngeal cancers were excluded from the analyses due to the low number of cases). The percentage of HPV-driven cases (considering HPV-E6*I mRNA positivity) was 1.8, 2.2, and 40.4% for OC, LC, and OPC cases, respectively (Table 1). The percentage of HPV-DNA/p16^INK4a^ double positivity was 2.4, 1.8, and 43.9% for OC, LC, and OPC cases, respectively. All OPC HPV-DNA-negative cases tested for p16^INK4a^ were also p16^INK4a^-negative, but three (15.0%) OC and one (3.8%) LC HPV-DNA-negative cases were p16^INK4a^-positive.

HPV16 was the most common type among HPV-DNA-positive cases for all HN sites, although with lower proportions in LC (59.3%) than in OC (90.0%) and OPC (96.4%) (Figure 1). The next most common HPV type was HPV18 for OC (10%) and LC (14.8%) and HPV33 (1.8%) for OPC. When only considering cases that are double-positive for HPV-DNA/p16^INK4a^, the prevalence of HPV16 increased in LC and OPC but not in OC. When only considering HPV-E6*I mRNA-positive cases, the prevalence of HPV16 increased in OC and OPC, but not in LC. Differences in HPV type distribution by HPV relatedness definitions were statistically significant. Appendix A show the demographic and clinical characteristics of the HNSCC cases included in the analysis, as well as the HPV prevalence and ORs for each biomarker for HPV positivity. Patients were mostly male (60.0% of OC, 78.0% of OPC, and 88.5% of LC), current or previous smokers (57.0% of OC, 67.9% of OPC, and 97.0% of LC), and current or previous drinkers (52.7% of OC, 60.6% of OPC, and 70.0% of LC). OPC patients most commonly had a locally advanced non-keratinizing SCC, whereas OC and LC had most commonly a locally advanced keratinizing grade 1 SCC and an early stage (I or II) keratinizing grade 2 SCC, respectively. Being a non-smoker or former smoker or diagnosed at more recent calendar periods were associated with HPV positivity in OPC for all three HPV-relatedness definitions. Younger ages (17–54 y) were associated with HPV-DNA positivity in OPC, but this association was not observed for HPV-DNA/p16^INK4a^ double positivity and HPV-E6*I mRNA positivity after adjusting for significant covariates such as tobacco use. Being younger was associated with HPV-DNA/p16^INK4a^ double positivity and HPV-E6*I mRNA positivity in LC. Being a non-smoker was associated with HPV-E6*I mRNA positivity in LC. None of these associations neither others were observed for OC after adjusting for significant covariates, with the exception of an anatomical location proximal to the oropharynx for HPV-DNA positivity. This association was not observed for HPV-DNA/p16^INK4a^ double positivity and HPV- E6*I mRNA positivity.

### 2.2. Overall and Progression-Free Survival of HPV-Driven OPC, OC, LC

HPV-driven OPC, but not HPV-driven OC or LC, showed better 5 year overall survival (OS) (*p* < 0.001), as compared to HPV-non-driven OPC (Figure 2). HPV-driven OPC also showed better 5 year progression-free survival (PFS) (*p* = 0.004), as compared to HPV-non-driven OPC (Figure 3). Both results were equivalent for HPV-DNA-positive and HPV-DNA/p16^INK4a^-double-positive cases. Other co-variates found to have a prognostic value for OS and PFS in univariate Cox proportional hazards models are shown in Table 2 and Table 3. Age was a prognostic factor for death for OPC and LC cases and for recurrence for all HNSCC. LC cases located distal to the oropharynx showed statistically significant improved OS and PFS. Clinical variables such as more advanced stages (7th TNM edition), node status > 1, multimodal treatment including surgery and positive margins for patients treated with surgery were also prognostic factors for death and recurrence for OC and LC, but not for OPC. Statistically significant improved OS among patients diagnosed in 2004–2007 was observed in OPC.

## 3. Discussion

To our knowledge, this study is the largest and most comprehensive assessment of the role of HPV in HNSCC in Italy. HPV-induced carcinogenesis is mediated by the oncoviral proteins E6 and E7, which, respectively, promote the degradation of the cellular proteins p53 and Rb, leading to cell proliferation, evasion from apoptosis, immortalization, and an increase of genomic instability [27]. The importance of determining whether an HN tumour is truly HPV-driven is underscored by the notion that these cancers are currently classified into two subtypes that must be considered as distinct entities: HPV-negative and HPV-positive. HPV-positive tumours, compared to the HPV-negative ones, are characterized by multiple molecular and clinicopathological differences, including age, socioeconomic status, prognosis, genetic landscape, and tissue differentiation [28,29]. Nevertheless, patients are treated with the same therapeutic protocols consisting mainly of surgery, radiation, and platinum-based chemotherapy [30].

Herein, we assess the prognostic value of different HPV-relatedness definitions, as well as factors related to HPV positivity in a retrospective cohort of approximately 700 HNSCC patients. We estimated that 1.8% of OC, 2.2% of LC, and 40.4% of OPC were HPV-driven based on HPV-RNA detection. The results were similar for HPV-DNA/ p16^INK4a^ double positivity.

A previous study of 248 HNC cases diagnosed in 2003–2012 in Northern Italy found HPV prevalence of 1.6% in OC, 20% in OPC, and 1% in LC, when considering HPV-DNA/HPV-E6*I RNA double positivity [10]. Our estimates were equivalent for OC and LC but considerably higher for OPC. Another study of 195 OPC cases diagnosed between 1992 and 2015 in Pavia, Italy, found a HPV prevalence of 37.9% when considering HPV-DNA/p16^INK4a^ double positivity [6], similar to our estimate of HPV-DNA/p16^INK4a^ double positivity in OPC, 43.9%. The study observed a marginally non-significant increase of HPV prevalence among OPC cases diagnosed after 2010 (45 vs. 28.3%, *p* = 0.06) [6]. A more recent Italian study also evaluated the role of HPV in patients with newly diagnosed OPC during the period 2000–2018, reporting a prevalence of HPV-driven OPC of 32.3% and a higher prevalence in the most recent years [11]. We also observed that OPC cases diagnosed at more recent periods (2008–2010) were independently associated with HPV positivity for all three HPV-relatedness definitions herein considered.

A cancer registry-based Italian study assessing the incidence and survival patterns of HNC diagnosed in Italy between 1988 and 2012 [3] found increasing incidence rates of OPC. These results, together with ours and those from others [6,11], suggest that HPV-related OPCs are increasing in Italy, as has previously happened in areas where nowadays most OPC cases are HPV-related. Regarding non-OPC sites, we did not observe any association between HPV positivity and calendar period.

Our study was conducted in the context of the international HPV-AHEAD study [31,32,33], which aimed to perform a comprehensive analysis of approximately 8000 HNC cases from Europe and India. A standardized protocol for optimizing the use of FFPE tissue blocks was developed, and each assay was performed in a single laboratory [33]. The first results on 355 Indian cases showed results equivalent to ours for HPV-E6*I mRNA positivity in LC (1.7%) and OC (1.6%), but considerably lower for OPC (9.4%) [32]. In a comprehensive study of 3680 HNC patients from 29 different countries, geographic heterogeneity of HPV-attributable fractions (HPV-AFs) was particularly evident for OPC [7].

HPV16 was the most common type among HPV-positive cases across all HN sites and HPV-relatedness definitions. However, its predominance was far higher in OC and OPC than in LC. Moreover, when considering only cases were HPV was the truly triggering carcinogenic agent (i.e., cases HPV-E6*I mRNA-positive) the percentage of HPV16-positive cases increased for OC and OPC but decreased for LC. These results, if confirmed in other larger studies, may have implications on the estimation of HPV vaccination effects.

Being a non-smoker or former smoker was associated with HPV positivity in OPC, consistently with what is reported in the literature regardless of geographical region [25,34,35]. However, as already observed in other studies [25], we did not find any association between gender and HPV positivity in OPC. HPV-driven OPC has been consistently associated with males in US studies [35], whereas in other regions, it has been associated with females [7]. These results highlight the geographical variability of the disease and the need of country-specific population-based studies to assess possible gender differences in HPV-driven OPC carcinogenesis.

Regarding non-oropharyngeal HNSCC cases, being younger and a non-smoker were associated with HPV positivity in LC. As compared to OPC, fewer studies have analysed the factors associated with HPV positivity in non-oropharyngeal cancer sites. The ICO international study also observed that HPV-positive LC patients, as well as OC ones, showed younger ages at diagnosis than HPV-negative ones [7]. However, we did not observe this association for OC after adjusting for co-variates.

As expected, HPV-positive OPC patients showed better OS and PFS than HPV-negative OPC cases regardless of the HPV-relatedness definition herein considered. However, as it has been already observed [23,36] the prognostic advantage of HPV-positive cases was higher when considering as positive only OPC cases truly driven by HPV infection (i.e., positive for HPV-E6*I mRNA or double-positive for HPV-DNA/p16^INK4a^). HPV-positive LC and OC cases did not show better OS and PFS than HPV-negative cases. Noteworthy, HPV-DNA/p16^INK4a^-positive and HPV-E6*I mRNA double-positive OC cases showed a marginally non-significant better PFS than the rest of cases. The prognostic value of HPV in non-oropharyngeal HNC is still unclear. While some studies have observed a better outcome for HPV-positive HNC, others have not [24]. The most updated guidelines for HPV testing in HNC have been published by the College of American Pathologists [24]. In this context, a panel of experts conducted a systematic review of studies that investigated the clinical outcomes of HPV-positive HNSCC. The panel concluded that there is no proven prognostic difference based on the presence or absence of HPV in non-oropharyngeal cancer. Thus, there is a need for more meta-analytical work to establish survival differences by HPV-relatedness of non-oropharyngeal cancer separated by anatomic site. The HPV AHEAD studies will contribute a considerable contribution to such pooled analyses.

We acknowledge that our study has several limitations. The retrospective nature of the cohort may have limited the accuracy of data related to some risk factors such as tobacco/alcohol use. Only 11% of HPV-DNA-negative cases were further tested for HPV-E6*I mRNA and p16^INK4a^, in accordance with the protocol established within the HPV-AHEAD consortium. For an important number of primary HNC cases consecutively diagnosed at IEO during the study period and targeted to be included in the study, no FFPE tissue blocks were available. Moreover, some differences were observed between cases with and without tissue sample, as it has been noted in other studies [25]. We had a small number of HPV-positive cases, and multivariate Cox’s proportional hazards models could not be performed due to the small number of deaths and recurrences. The small number of cases also hampered the performance of Kaplan–Meier analyses on locally advanced cases only. However, stage (according to 7th edition TNM) was not found to be associated with HPV positivity. For non-OPC cases, the small number of HPV-positive cases has made it difficult to extrapolate meaning from the survival analyses and results must be taken with caution. HPV-AHEAD definition of p16^INK4a^ positivity was established before the publication of the guidelines for HPV testing in HNC by the College of American Pathologists [24], where a 70% nuclear and cytoplasmic staining cut-off was recommended and is currently widely accepted in clinical practice. However, the impact of misclassification can be considered low due to the low number of p16^INK4a^-positive cases, which are less than 70%. A total of 15 out of 68 cases classified as p16^INK4a^-positive did not reach the 70% nuclear and cytoplasmic staining threshold.

## 4. Materials and Methods

### 4.1. Study Design and Samples

A retrospective cohort of patients consecutively diagnosed with a primary HNC at the IEO in Milan from 2000 to 2010 was conducted within the HPV-AHEAD project [31,32,33]. HNC cases were identified from medical records/pathology reports of the hospital. Selected cases had a histopathological diagnosis of primary squamous cell carcinoma of the oropharynx (International Classification of Diseases for Oncology (ICD-O) C01.9, C02.4, C05.1, C05.2, C09, C10), oral cavity (ICD-O: C02.0–C06.9, excluding C02.4, C05.1, C05.2), the hypopharynx, and larynx (ICD-O: C13, C32).

Information on demographics, smoking and alcohol consumption, and clinical and follow-up data up to 2019 was extracted from electronic medical records. The definition of a drinker was consumption of three or more drinks per week. We only considered FFPE tumour samples from the diagnosis previous to treatment.

Ethical clearance was obtained from IEO Ethical Committee (code IEO N101), Milan, Italy as well as IARC, Lyon, France. The study did not involve any contact with the patients. All clinical and pathological data were collected using well-designed case report forms (CRF) according to good clinical practice guidelines.

Adequate measures to ensure data protection, confidentiality, patients’ privacy, and anonymization were taken into account. FFPE tissue blocks were used to perform several laboratory assays, as described below. Each assay was performed in a single laboratory: (i) HPV-DNA assay at IARC, (ii) HPV RNA assay at DKFZ, and (iii) p16^INK4a^ staining at Roche mtm laboratories.

Cancer samples having tested negative for both HPV-DNA and beta-globin DNA were excluded from the analyses.

### 4.2. Preparation of the Tissue Sections

FFPE tissues were in part processed at the International Agency for Research on Cancer, Lyon, France and at IEO, Milan, Italy, following the HPV-AHEAD sectioning protocol [33].

### 4.3. Histological Analysis

All sections were evaluated by the HPV-AHEAD histopathology review panel six pathologists (J.P.B., B.L.R., F.M., O.C., R.V.K., N.G.). An online pathology form was used. Each pathologist evaluated tissues from approximately 80 patients. All sections were re-evaluated by a second panel of pathologists. Only FFPE tissue blocks for which the first and last haematoxylin/eosin stained sections reflected tumour tissue were included in the study.

### 4.4. HPV-DNA Genotyping

DNA was purified from three consecutive FFPE sections (S6–8), as previously described [31]. HPV-DNA was detected by E7 type-specific multiplex genotyping (E7-MPG), which combines multiplex polymerase chain reaction (PCR) and hybridization to type-specific oligonucleotide probes on fluorescent beads (Luminex Corporation, Austin, TX) [37,38]. TS-MPG uses HPV type-specific primers targeting the E7 region of 19 high-risk (HR) or possible/probable HR (pHR) HPV types (HPV16, 18, 26, 31, 33, 35, 39, 45, 51, 52, 53, 56, 58, 59, 66, 68a and b, 70, 73, and 82) and two low-risk (LR) HPV types (HPV6 and 11). Detection limits range from 10 to 1000 copies of the viral genome per reaction. Two primers for amplification of the beta-globin gene were also included to control for the quality of the template DNA. A slightly modified E7-MPG with higher analytical sensitivity was performed with amplicon size of approximately 100 bp for 10 HPV types (HPV16, 18, 31, 33, 35, 52, 56, 66, 6, and 11), and 117 bp for beta-globin [20,39]. After PCR amplification, 10 µl of each reaction mixture were analysed by multiplex HPV genotyping (MPG) using Luminex technology (Luminex Corporation) as described previously [40].

All HPV-DNA-positive FFPE specimens and a random subgroup of approximately 11% of HPV-DNA-negative cases were subjected to HPV-E6*I mRNA detection and p16^INK4a^ staining.

### 4.5. HPV E6*I RNA Analysis

An ultra-short amplicon, E6*I mRNA RT-PCR assay was chosen for HPV-mRNA detection for its applicability to FFPE material and absolute RNA specificity by using a splice-site as identification target [20]. Total RNA was purified from three pooled consecutive sections of the same tissue block using the PureLink FFPE Total RNA Isolation Kit (Invitrogen, Carlsbad, CA) [20]. The HPV type-specific E6*I mRNA assays are available for 20 HR- or pHR-HPV types for which presence of splice sites was demonstrated. Briefly, extracted RNA was subjected to a one-step reverse transcription PCR protocol with the QuantiTect Virus Kit (Qiagen, Hilden, Germany) using HPV type specific primers to amplify 65–75 bp cDNA sequences across the E6*I splice sites and human ubiquitin C (ubC) primers that generate a 85 bp cDNA amplicon as a control for tissue and RNA quality. The products were then hybridized to HPV and ubC specific oligonucleotide probes coupled to fluorescence-labelled Luminex beads (Luminex Corp.) as described previously [20]. Specimens that were HPV-E6*I and/or ubC mRNA-positive were considered RNA valid.

### 4.6. p16^INK4a^ IHC

Expression of p16 was evaluated manually by IHC in FFPE sections using the CINtec p16 Histology Kit (Roche mtm laboratories AG, Mannheim, Germany) according to the manufacturer’s instructions and as previously described [32]. A continuous, diffuse staining for p16^INK4a^ within the cancer area of the tissue sections was considered as positive, while a focal staining or no staining was considered negative. IHC slides were analysed without knowledge of any other clinical information (including HPV-DNA and RNA status) by R.R. and F.M. Discordant cases were analysed by O.C. and consensus was reached.

### 4.7. Statistical Analyses

Differences in the covariate distribution (age, gender, sub-site, tobacco and alcohol use, TNM 7th Edition) between cases with and without available FFPE block were assessed by Pearson’s chi2 test. We used HPV-E6*I mRNA positivity as the reference test for viral carcinogenic activity. We assumed that HPV-DNA-negative cases not tested for HPV-E6*I mRNA were HPV-E6*I mRNA-negative and HPV-DNA/HPV-E6*I mRNA double-positive cases were considered as HPV-E6*I mRNA-positive cases. HPV-prevalence and HPV-type distribution were assessed. Unconditional logistic regression analyses were performed to identify independent factors (i.e., age, sex, tobacco-alcohol use, clinical data) associated with HPV etiological involvement in each HNSCC site according to the three different HPV-relatedness definitions using backward selection of significant covariates. The likelihood ratio test (LR *p* < 0.05) and the Akaike information criterion were applied to exclude non-significant factors. Histological variables were not considered in multivariate analyses as they were considered to be intermediate variables in the carcinogenic process, as previously described [41]. Crude and adjusted ORs and their 95% confidence intervals (CI) were estimated. Survival time was defined as the period between the date of histological diagnosis and the date of death for any cause (OS) or the date of cancer recurrence (PFS). The cumulative probability of survival was estimated by Kaplan–Meier analyses. Survival curves were compared with the log-rank test for each HPV-relatedness definition and each head and neck site up to 5 years. Univariate Cox proportional hazards models were also conducted up to 5 years to assess the prognostic role of HPV status and other co-variates. Multivariate Cox proportional hazards models could not be performed due to the small number of deaths and recurrences. Metastatic patients (stage IVc, 7th edition TNM) were excluded from survival analyses. Statistical significance for all analyses was set at the 2-sided 0.05 level. Data analyses were performed with STATA software v.16 (Stata Corp., College Station, TX, USA).

## 5. Conclusions

Our findings from a large cohort of unselected HNSCC patients provide a comprehensive picture on the role of HPV in OPC and non-oropharyngeal cancer in a setting of Southern Europe, which is a region currently considered to bear a low burden of HPV-driven HNC. We have observed an increase of the prevalence of HPV positivity in OPC with most recent calendar periods. These results, together from previous findings of increasing incidence of OPC in Italy [3], suggest that HPV-driven OPC is increasing in Italy, as has happened before in other developed regions. However, we observed some differences with respect to published data from North America. Our results also suggest that current estimations of HPV prevalence in Southern Europe are outdated and warrant updated population-based studies. Moreover, our study provides novel insights on the type-specific contribution of HPV, not only on OPC but also on LC and OC, that may have implications when estimating the possible protective effect of HPV vaccination against HNC.

## Figures and Tables

**Figure 1 cancers-12-03567-f001:**
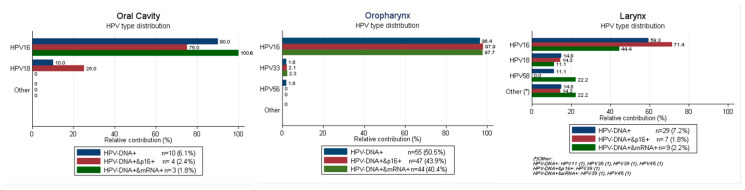
HPV type distribution in HPV-driven HN sites according to different combination of biomarkers: HPV-DNA, HPV-DNA/p16^INK4a^, and HPV-DNA/E6*I mRNA detection.

**Figure 2 cancers-12-03567-f002:**
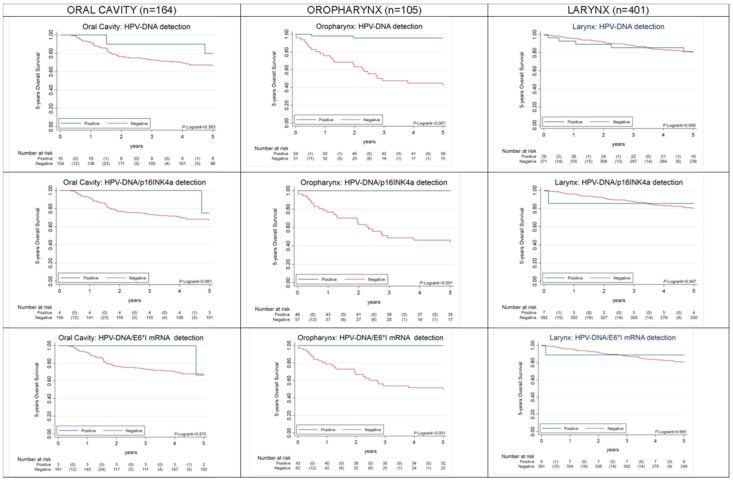
Five-year overall survival of OPC, OC, LC patients (stage IVc patients are excluded) by HPV positivity according to three different HPV-relatedness definitions: HPV-DNA, HPV-DNA/p16^INK4a^, and HPV-DNA/E6*I mRNA detection.

**Figure 3 cancers-12-03567-f003:**
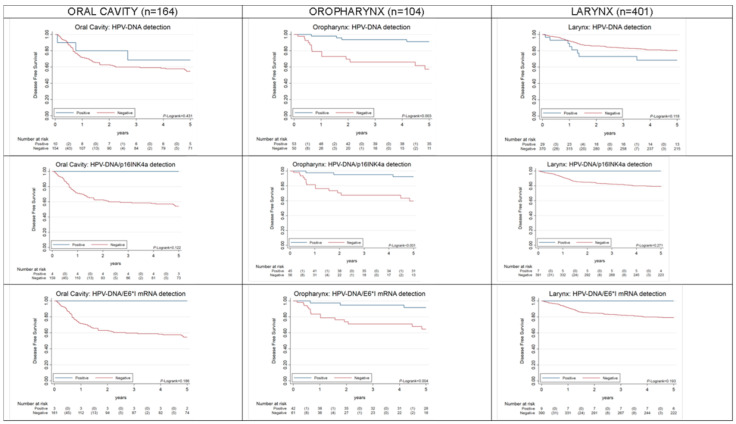
Five-year progression-free survival of OPC, OC, LC patients (stage IVc patients are excluded) by HPV positivity according to three different HPV-relatedness definitions: HPV-DNA, HPV-DNA/p16^INK4a^, and HPV-DNA/E6*I mRNA detection.

**Table 1 cancers-12-03567-t001:** Human papillomavirus (HPV)-attributable fractions by different combinations of biomarkers: HPV-DNA, p16^INK4a^ and E6*I mRNA detection.

Total HNC (*n* = 675)	Oral Cavity (OC)		Oropharynx (OPC)		Larynx (LC)	
	*n* = 165 (23.7%) ^a^		*n* = 109 (15.7%) ^a^		*n* = 401 (57.6%) ^a^	
	*n*	*%* **^b^**	*n*	*%* **^b^**	*n*	*%* **^b^**
**HPV markers positivity**						
HPV-DNA+	10/165	6.1%	55/109	50.5%	29/401	7.2%
p16^INK4a^+ in HPV-DNA+ cases	4/9	44.4%	47/53	88.7%	7/28	25.0%
p16^INK4a^+ in HPV-DNA- cases	3/20	15.0%	0/8	0.0%	1/26	3.8%
E6*I mRNA+ in HPV-DNA+ cases	3/10	30.0%	44/55	80.0%	9/29	31.0%
E6*I mRNA+ in HPV-DNA- cases	0/21	0.0%	0/8	0.0%	0/33	0.0%
HPV-DNA+ AND p16^INK4a^+	4/164	2.4%	47/107	43.9%	7/400	1.8%
HPV-DNA+ AND E6*I mRNA+	3/165	1.8%	44/109	40.4%	9/401	2.2%
HPV-DNA+ AND [E6*I mRNA+ OR p16^INK4a^+]	4/164	2.4%	47/107	43.9%	9/400	2.3%
HPV-DNA+ AND E6*I mRNA+ AND p16^INK4a^+	3/164	1.8%	42/107	39.3%	6/400	1.5%

“HNC”: head and neck cancer; “OC”: oral cavity cancer; “OPC”: oropharyngeal cancer; “LC”: laryngeal cancer; HPV-DNA detection using a type-specific PCR bead-based multiplex genotyping (E7-MPG) assay that combines multiplex polymerase chain reaction (PCR) and bead-based Luminex technology (Luminex Corp., Austin, TX, USA); p16^INK4a^ considered as positive when a continuous, diffuse staining cells for p16^INK4a^ within the cancer area of the tissue sections was observed. Performed in HPV-DNA-positive cases and in a 11% of random HPV-DNA-negative cases; E6*I mRNA performed in case of HPV-DNA-positive cases for any of the 20 high-risk genotypes detectable by the technique. ^a^: % of invasive cancer cases included for each anatomic head and neck (HN) sublocation among all HNC included in the study. ^b^: % of positive cases for each combination of HPV-biomarker results among total cases analysed for each HPV-biomarker.

**Table 2 cancers-12-03567-t002:** Hazard ratios and 95% CI for death for OPC, OC, LC patients (stage IVc patients are excluded) for 5 year follow-up.

Covariate	OPC	Crude HR				OC	Crude HR				LC	Crude HR			
	Cases/Deaths	HR	95% CI		*p*-Value	Cases/Deaths	HR	95% CI		*p*-Value	Cases/Deaths	HR	95% CI		*p*-Value
			Lower	Upper				Lower	Upper				Lower	Upper	
**HPV-DNA**	**105/26**				**0.000**	**164/51**				0.322	**401/69**				0.990	
Other	51/24	Ref.				154/49	Ref.				372/64	Ref.			
HPV-DNA+	54/2	**0.05**	**0.01**	**0.22**		10/2	0.52	0.13	2.16		29/5	1.01	0.40	2.50	
**HPV-E6*I mRNA**	**105/26**				**-**	**164/51**				0.872	**401/69**				0.666
Other	62/26					161/50	Ref.				392/68	Ref.			
HPV-E6*I mRNA+	43/0	-	-	-		3/1	0.85	0.12	6.18		9/1	0.67	0.09	4.80	
**HPV-DNA and p16**	**103/26**				**-**	**163/50**				0.640	**400/69**				0.946
Other	57/26					159/49	Ref.				393/68	Ref.			
HPV-DNA+ and p16+	46/0	-		-	-	4/1	0.64	0.09	4.67		7/1	0.93	0.13	6.73	
**Age at diagnosis**	**105/26**	**1.05**	**1.01**	**1.10**	**0.023**	**164/51**	1.01	0.99	1.03	0.190	**401/69**	**1.04**	**1.01**	**1.07**	**0.002**
17–54 y	24/3	Ref.			0.076	71/17	Ref.			0.142	70/9	Ref.			0.205
55–62 y	38/8	1.79	0.46	6.76		28/12	2.30	1.10	4.82		115/20	1.42	0.65	3.12	
63–70 y	22/7	2.64	0.68	10.20		27/11	1.85	0.87	3.95		108/16	1.28	0.57	2.90	
71–94 y	21/8	4.84	1.28	18.31		38/11	1.38	0.65	2.94		108/24	2.10	0.98	4.52	
**Gender**	**105/26**				0.778	**164/51**				0.898	**401/69**				0.906
Male	82/21	Ref.				99/31	Ref.				355/61	Ref.			
Female	23/5	0.87	0.33	2.31		65/20	1.04	0.59	1.82		46/8	1.05	0.50	2.18	
**Period of diagnosis**	**105/26**	**0.87**	**0.76**	**0.99**	**0.028**	**164/51**	0.98	0.90	1.07	0.725	**401/69**	1.00	0.92	1.08	0.914
2000–2003	38/16	Ref.			**0.028**	49/15	Ref.			0.447	147/24	Ref.			0.612
2004–2007	46/7	**0.32**	**0.13**	**0.79**		68/24	1.28	0.67	2.44		170/33	1.23	0.73	2.08	
2008–2010	21/3	0.36	0.10	1.22		47/12	0.83	0.39	1.78		84/12	0.92	0.46	1.85	
**Tobacco use**	**105/26**				0.091	**164/51**				0.775	**401/69**				0.783
Non-smoker	21/3	Ref.				61/17	Ref.				11/2	Ref.			
Former smoker	21/3	1.04	0.21	5.14		31/11	1.22	0.57	2.61		28/6	1.28	0.26	6.34	
Smoker	51/18	2.84	0.84	9.65		63/21	1.21	0.64	2.29		361/61	0.89	0.22	3.66	
Unknown	12/2	0.99	0.17	5.95		9/2	0.65	0.15	2.83		1/0	-			
**Alcohol use**	**105/26**				0.276	**164/51**				0.089	**401/69**				0.166
Non-drinker	25/4	Ref.				66/15	Ref.				115/18	Ref.			
Former drinker	3/2	4.90	0.89	26.86		3/0	-				3/1	3.28	0.44	24.6	
Drinker	62/17	1.97	0.66	5.87		84/33	1.80	0.98	3.32		274/50	1.19	0.69	2.04	
Unknown	15/3	1.13	0.25	5.06		11/3	1.05	0.30	3.61		9/0	-			
**Subsite**	105/26				0.110					-					-
Tonsil	47/9	Ref.				-	-				-	-			
BOT	29/6	1.16	0.41	3.26		-	-				-	-			
Other oropharynx	29/11	2.50	1.03	6.03		-	-				-	-			
					-	**164/51**				0.437	**401/69**				**0.024**
Proximal to oropharynx	-	-				49/13	Ref.				97/23	Ref.			
Distal to oropharynx	-	-				115/38	1.28	0.68	2.39		304/46	**0.55**	**0.33**	**0.91**	
**Stage (7th edition TNM)**	**105/26**				0.852	**164/51**				**0.040**	**401/69**				**0.000**
I + II	24/6	Ref.				74/17	Ref.				256/30	Ref.			
III	22/5	0.80	0.24	2.61		28/9	1.58	0.70	3.55		77/13	1.53	0.80	2.94	
IVa + IVb	59/15	1.06	0.41	2.73		62/25	**2.19**	**1.18**	**4.07**		68/26	**4.05**	**2.39**	**6.85**	
**cN**	**105/26**				0.209	**164/51**				**0.007**	**401/69**				**0.000**
0	37/12	Ref.				94/24	Ref.				339/46	Ref.			
1	22/4	0.52	0.17	1.61		37/11	1.24	0.61	2.54		16/4	2.17	0.78	6.04	
2	39/7	0.52	0.20	1.32		33/16	**2.89**	**1.53**	**5.45**		41/15	**3.53**	**1.97**	**6.33**	
3	7/3	1.93	0.54	6.89		0					5/4	**7.64**	**2.74**	**21.28**	
**Treatment**	**105/26**				0.204	**164/51**				**0.000**	**401/69**				**0.025**
Only surgery	22/8	Ref.				85/17	Ref.				267/37	Ref.			
Surgery + others	41/11	0.53	0.21	1.33		71/29	2.54	1.39	4.62		107/27	2.03	1.24	3.34	
Conservative	37/6	0.36	0.13	1.05		3/1	1.87	0.25	14.04		19/3	1.71	0.53	5.56	
Unknown	5/1	1.96	0.23	16.51		5/4	25.57	8.17	80.10		8/2	3.93	0.95	16.32	
**Positive margins**	**77/18**				0.299	**159/49**				**0.041**	**353/68**				**0.039**
No	60/16	Ref.				144/42	Ref.				294/43	Ref.			
Yes	17/2	0.49	0.11	2.14		15/7	**2.54**	**1.14**	**5.68**		59/15	**1.92**	**1.07**	**3.46**	
**Time of follow-up**															
Median (years)	5.08					9.05					6.83				
(Min–Max)	(0.02–18.71)					(0.02–19.05)					(0.00–18.51)				
**Time since dead**															
Median (years)	1.97					1.97					3.43				
(Min–Max)	(0.02–16.21)					(0.02–16.21)					(0.01–16.40)				

95% CI: confidence interval; OPC: oropharyngeal cancer; OC: oral cavity cancer; LC: laryngeal cancer; HR: hazard ratio; cN: clinical node status; Tonsil: C02.4 and C09.0 and C09.1 and C09.9; BOT: base of the tongue (C01); other oropharynx: C10 and C10.0 and C10.2 and C10.3 and C10.8 and C10.9; “OC Distal to oropharynx:C02 and C02.0 and C02.1 0 and C02.2 and C02.3 and C03.1 and C04.1 and C04.9 and C06.0; OC Proximal to oropharynx: C02.8 and C02.9 and C05.8 and C06.2; LC Proximal to oropharynx: C32.1 and C32.8; LC Distal to oropharynx: C32.0; Std: standard deviation; Min: minimum; Max: maximum. Stage IVc patients are excluded. Bold represents statistically significant categories.

**Table 3 cancers-12-03567-t003:** Hazard ratios and 95% CI for recurrence for OPC, OC, LC patients (stage IVc patients are excluded) for 5 year follow-up.

Covariate	OPC	Crude HR				OC	Crude HR				LC		Crude HR		
	Cases/Rec.	HR	95% CI		*p*-Value	Cases/Rec.	HR	95% CI		*p*-Value	Cases/Rec.	HR	95% CI		*p*-Value
			Lower	Upper				Lower	Upper				Lower	Upper	
**HPV-DNA**	**105/20**				**0.000**	**164/70**				0.402	**401/75**				0.152
Other	51/15	Ref.				154/67	Ref.				372/67	Ref.			
HPV-DNA+	54/5	**0.16**	**0.05**	**0.50**		10/3	0.63	0.20	2.01		29/8	1.78	0.85	3.71	
**HPV-E6*I mRNA**	**105/20**				**0.003**	**164/70**					**401/75**				
Other	62/16	Ref.				161/70					392/75				
HPV-E6*I mRNA+	43/4	**0.19**	**0.06**	**0.67**		3/0	-				9/0	-			
**HPV-DNA and p16**	**103/20**				**0.001**	**163/70**				-	**400/75**				-
Other	57/16	Ref.				159/70					393/75				
HPV-DNA+ and p16+	46/4	**0.15**	**0.04**	**0.52**		4/0	-				7/0	-			
**Age at diagnosis**	**105/20**	**1.04**	**0.99**	**1.10**	**0.107**	**164/70**	1.02	1.00	1.03	0.013	**401/75**	**1.03**	**1.01**	**1.05**	**0.017**
17–54 y	24/2	Ref.			0.097	71/25	Ref.			0.079	70/10	Ref.			0.202
55–62 y	38/7	2.21	0.45	10.94		28/16	2.08	1.11	3.90		115/18	1.16	0.54	2.52	
63–70 y	22/8	5.45	1.16	25.70		27/10	1.10	0.53	2.29		108/25	1.85	0.88	3.86	
71–94 y	21/3	2.04	0.29	14.51		38/19	1.78	0.98	3.23		108/22	1.77	0.84	3.73	
**Gender**	**105/20**				0.364	**164/70**				0.404	**401/75**				0.867
Male	82/13	Ref.				99/40	Ref.				355/67	Ref.			
Female	23/7	1.65	0.59	4.62		65/30	1.22	0.76	1.97		46/8	0.94	0.45	1.96	
**Period of diagnosis**	**105/20**	0.92	0.79	1.07	0.270	**164/70**	1.01	0.94	1.09	0.698	**401/75**	1.02	0.94	1.10	0.636
2000–2003	38/9	Ref.			0.312	49/21	Ref.			0.986	147/26	Ref.			0.799
2004–2007	46/6	0.45	0.16	1.27		68/28	1.05	0.59	1.84		170/31	1.06	0.63	1.79	
2008–2010	21/5	0.65	0.18	2.40		47/21	1.04	0.57	1.90		84/18	1.23	0.67	2.27	
**Tobacco use**	**105/20**				0.168	**164/70**				0.845	**401/75**				0.395
Non-smoker	21/5	Ref.				61/27	Ref.				11/2	Ref.			
Former smoker	21/1	0.29	0.03	2.80		31/11	0.76	0.38	1.54		28/7	1.62	0.34	7.81	
Smoker	51/12	1.69	0.48	6.00		63/27	0.93	0.55	1.59		361/66	0.98	0.24	4.01	
Unknown	12/2	0.82	0.14	4.92		9/5	1.16	0.45	3.02		1/0	-			
**Alcohol use**	**105/20**				0.388	**164/70**				0.829	**401/75**				0.062
Non-drinker	25/5	Ref.				66/26	Ref.				115/20	Ref.			
Former drinker	3/1	2.12	0.25	18.30		3/2	1.84	0.44	7.76		3/0	-			
Drinker	62/13	0.99	0.34	2.86		84/37	1.18	0.71	1.95		274/55	1.16	0.69	1.93	
Unknown	15/1	0.27	0.03	2.28		11/5	1.04	0.40	2.71		9/0	-			
**Subsite**	105/20				0.549										
Tonsil	47/10	Ref.													
BOT	29/4	0.81	0.24	2.68											
Other oropharynx	29/6	1.59	0.55	4.59											
**Subsite**						**164/70**				0.258	**401/75**				**0.012**
Proximal to Opx						49/17	Ref.				97/26	Ref.			
Distal to Opx						115/53	1.36	0.79	2.35		304/49	**0.54**	**0.33**	**0.87**	
**Stage (7th edition TNM)**	**105/20**				0.497	**164/70**				0.473	**401/75**				**0.001**
I + II	24/7	Ref.				74/33	Ref.				256/36	Ref.			
III	22/4	0.60	0.17	2.14		28/9	0.67	0.32	1.41		77/16	1.47	0.81	2.69	
IVa + IVb	59/9	0.52	0.18	1.51		62/28	1.04	0.63	1.71		68/23	**2.94**	**1.74**	**4.97**	
**cN**	**105/20**				0.111	**164/70**				**0.033**	**401/75**				**0.001**
0	37/8	Ref.				94/40	Ref.				339/50	Ref.			
1	22/4	0.90	0.26	3.08		37/11	0.64	0.33	1.25		16/7	**3.18**	**1.36**	**7.042**	
2	39/5	0.48	0.14	1.65		33/19	1.68	0.97	2.91		41/16	**3.44**	**1.96**	**6.05**	
3	7/3	3.83	0.97	14.98		0					5/2	3.95	0.96	16.26	
**Treatment**	**105/20**				0.090	**164/70**				0.699	**401/75**				0.052
Only surgery	22/6	Ref.				85/35	Ref.				267/45	Ref.			
Surgery + others	41/5	0.23	0.06	0.80		71/32	1.09	0.67	1.76		107/24	1.47	0.89	2.42	
Conservative treatment	37/9	0.65	0.22	1.86		3/1	0.72	0.10	5.26		19/6	3.00	1.27	7.05	
Unknown	5/0	-				5/2	2.50	0.59	10.51		8/0	-			
**Positive margins**	**77/13**				0.189	**159/69**				**0.014**	**353/65**				**0.039**
No	60/12	Ref.				144/59	Ref.				294/49	Ref.			
Yes	17/1	0.31	0.04	2.41		15/10	**2.58**	**1.31**	**5.06**		59/16	**1.87**	**1.07**	**3.31**	

Rec.: recurrences; 95% CI: confidence interval; OPC: oropharyngeal cancer; OC: oral cavity cancer; LC: laryngeal cancer; HR: hazard ratio; cN: clinical node status; Tonsil: C02.4 and C09.0 and C09.1 and C09.9; BOT: base of the tongue (C01); other oropharynx: C10 and C10.0 and C10.2 and C10.3 and C10.8 and C10.9; “OC Distal to oropharynx:C02 and C02.0 and C02.1 0 and C02.2 and C02.3 and C03.1 and C04.1 and C04.9 and C06.0; OC Proximal to oropharynx: C02.8 and C02.9 and C05.8 and C06.2; LC Proximal to oropharynx: C32.1 and C32.8; LC Distal to oropharynx: C32.0; Std: standard deviation; Min: minimum; Max: maximum. Stage IVc patients are excluded. Bold represents statistically significant categories.

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
