# Peer review of "Role of Human Papillomavirus Infection in Head and Neck Cancer in Italy: The HPV-AHEAD Study"

_cancers, 2020, doi:10.3390/cancers12123567_

Round 1

Reviewer 1 Report

This is an interesting study about the role of Human Papillomavirus infection in Head and Neck Cancer. p16INK4a, HPV DNA and HPV 57 RNA biomarkers were used to assess the HPV status in Head and Neck Cancer in a retrospective cohort of approximately 700 patients. The authors found that HPV prevalence in oropharyngeal cancers was higher than in oral and laryngeal cancers and HPV positivity conferred better prognosis only in oropharyngeal tumors.

The paper is well written. However, some issues remain.

At lines 159-161 the authors stated that “All OPC HPV-159 DNA negative cases tested for HPV-E6*I mRNA were also HPV-E6*I mRNA negative, but 4 (21.1%) 160 OC and 3 (9.4%) LC HPV-DNA negative cases were p16INK4a positive.” There is probably a mistake in the sentence. Please correct it.

In the Materials and Methods section, the authors should report the definition of drinker that was used in the study.

In Table 2, measurements units for time of follow-up and time since death are lacking.

Type of treatment and other factors, such as margins and node status, can influence prognosis. Therefore, they should be reported and, if possible, included in the survival analyses.

A brief description of different molecular patterns of HPV-negative and HPV-positive head and neck carcinomas may be helpful in the Discussion section.

Author Response

20-11-2020

Response to reviewers

Thank you very much for having reviewed our work. Please find below our answers and explanations on how we have addressed the Reviewers’ comments, point by point, in order to proceed with the submission of the revised version of the manuscript to Cancers. A new version of the manuscript, with track changes and clean, has been also prepared and added to the submission.

Reviewer no 1

(x) I would not like to sign my review report

( ) I would like to sign my review report

English language and style

( ) Extensive editing of English language and style required

( ) Moderate English changes required

(x) English language and style are fine/minor spell check required

( ) I don't feel qualified to judge about the English language and style

          Yes    Can be improved     Must be improved    Not applicable

Does the introduction provide sufficient background and include all relevant references?

          (x)      ( )       ( )       ( )

Is the research design appropriate?

          (x)      ( )       ( )       ( )

Are the methods adequately described?

          ( )       (x)      ( )       ( )

Are the results clearly presented?

          ( )       (x)      ( )       ( )

Are the conclusions supported by the results?

          (x)      ( )       ( )       ( )

Comments and Suggestions for Authors

This is an interesting study about the role of Human Papillomavirus infection in Head and Neck Cancer. p16INK4a, HPV DNA and HPV 57 RNA biomarkers were used to assess the HPV status in Head and Neck Cancer in a retrospective cohort of approximately 700 patients. The authors found that HPV prevalence in oropharyngeal cancers was higher than in oral and laryngeal cancers and HPV positivity conferred better prognosis only in oropharyngeal tumors.

The paper is well written. However, some issues remain.

At lines 159-161 the authors stated that “All OPC HPV-159 DNA negative cases tested for HPV-E6*I mRNA were also HPV-E6*I mRNA negative, but 4 (21.1%) 160 OC and 3 (9.4%) LC HPV-DNA negative cases were p16INK4a positive.” There is probably a mistake in the sentence. Please correct it.

Thank you very much for noting this. There was indeed a mistake in the sentence which has been corrected in the following way:

“All OPC HPV-DNA negative cases tested for p16INK4a were also p16INK4a negative, but 3 (15.0%) OC and 1 (3.8%) LC HPV-DNA negative cases were p16INK4a positive.”

In the Materials and Methods section, the authors should report the definition of drinker that was used in the study.

The definition of drinker that was used in the study has been reported in the materials and methods section as following:

“The definition of drinker was consumption of three or more drinks per week.”

In Table 2, measurements units for time of follow-up and time since death are lacking.

The measurements units are years. This has been indicated for time of follow-up and time since death in table 2.

Type of treatment and other factors, such as margins and node status, can influence prognosis. Therefore, they should be reported and, if possible, included in the survival analyses.

Hazard ratios for death and recurrence by type of treatment, margins and node status have been estimated and presented in Tables 2 and 3.

A brief description of different molecular patterns of HPV-negative and HPV-positive head and neck carcinomas may be helpful in the Discussion section.

The following paragraph has been added in the discussion section:

“HPV-induced carcinogenesis is mediated by the oncoviral proteins E6 and E7, which, respectively, promote the degradation of the cellular proteins p53 and Rb, leading to cell proliferation, evasion from apoptosis, immortalization, and an increase of genomic instability [27]. The importance of determining whether an HN tumour is truly HPV-driven is underscored by the notion that these cancers are currently classified into two subtypes that must be considered as distinct entities: HPV-negative and the HPV-positive tumours. HPV positive tumors, compared to the HPV negative ones, are characterized by multiple molecular and clinicopathological differences, including age, socioeconomic status, prognosis, genetic landscape and tissue differentiation [28,29]. Nevertheless, patients are treated with the same therapeutic protocols consisting mainly in surgery, radiation and platinum-based chemotherapy [30].”

Reviewer 2 Report

Tagliabue et al. report the results of a large and comprehensive retrospective cohort study on human papillomavirus and head and neck cancers in Italy. Similar studies with similar outcomes in other settings have been published but this is still an important contribution to the field.

  1. The main issue with the data is that there are extremely small numbers of HPV-positive oral cavity cancer cases versus HPV-negative cases, which makes it difficult to extrapolate meaning from the Kaplan-Meier plots. This is also true for laryngeal cancers analysed by HPV/p16 and HPV/E6*I mRNA. Thus, only the oropharyngeal data are sufficiently robust to allow firm conclusions to be made.
  2. In Results section 2.2 the authors should expand significantly their description on co-variates to bring out the other pieces of information from the graphs and tables.
  3. Why did the authors not consider HPV+ve/p16/E6*I positivity as a triple biomarker? Surely this would be the most stringent analysis of HPV activity in the clinical samples.
  4. For the less specialist reader, the authors should explain clearly why they used E6*I mRNA, rather than E6 itself detection in their analysis.

Author Response

20-11-2020

Response to reviewers

Thank you very much for having reviewed our work. Please find below our answers and explanations on how we have addressed the Reviewers’ comments, point by point, in order to proceed with the submission of the revised version of the manuscript to Cancers. A new version of the manuscript, with track changes and clean, has been also prepared and added to the submission.

Reviewer no.2

(x) I would not like to sign my review report

( ) I would like to sign my review report

English language and style

( ) Extensive editing of English language and style required

( ) Moderate English changes required

(x) English language and style are fine/minor spell check required

( ) I don’t feel qualified to judge about the English language and style

          Yes    Can be improved     Must be improved    Not applicable

Does the introduction provide sufficient background and include all relevant references?

          (x)      ( )       ( )       ( )

Is the research design appropriate?

          (x)      ( )       ( )       ( )

Are the methods adequately described?

          (x)      ( )       ( )       ( )

Are the results clearly presented?

          ( )       (x)      ( )       ( )

Are the conclusions supported by the results?

          ( )       (x)      ( )       ( )

Comments and Suggestions for Authors

Tagliabue et al. report the results of a large and comprehensive retrospective cohort study on human papillomavirus and head and neck cancers in Italy. Similar studies with similar outcomes in other settings have been published but this is still an important contribution to the field.

    The main issue with the data is that there are extremely small numbers of HPV-positive oral cavity cancer cases versus HPV-negative cases, which makes it difficult to extrapolate meaning from the Kaplan-Meier plots. This is also true for laryngeal cancers analysed by HPV/p16 and HPV/E6*I mRNA. Thus, only the oropharyngeal data are sufficiently robust to allow firm conclusions to be made.

This Reviewer is correct. This has now been more emphasized in the limitations section, as follows:

“For non-OPC cases, the small number of HPV-positive cases has made it difficult to extrapolate meaning from the survival analyses and results must be taken with caution.”

    In Results section 2.2 the authors should expand significantly their description on co-variates to bring out the other pieces of information from the graphs and tables.

The description of the results from survival analyses of the different co-variates evaluated has been described in more detail in the text as follows:

“Age was a prognostic factor for death for OPC and LC cases, and for recurrence for all HNSCC. LC cases located distal to the oropharynx showed statistically significant improved OS and PFS. Clinical variables such as more advanced stages (7th TNM edition), node status > 1, multimodal treatment including surgery and positive margins for patients treated with surgery were also prognostic factors for death and recurrence for OC and LC, but not for OPC. Statistically significant improved OS among patients diagnosed at 2004-2007 was observed in OPC.”

Why did the authors not consider HPV+ve/p16/E6*I positivity as a triple biomarker? Surely this would be the most stringent analysis of HPV activity in the clinical samples.

We did not present estimates of HPV+ve/p16/E6*I triple positivity because we considered that the comparison between HPV-DNA/p16 double positivity and the positivity for the Gold-Standard for HPV causality, i.e. E6*I, was of most clinical interest, especially in non-OPC cases were p16 use for HPV diagnosis is not currently recommended. Moreover, there is a need for diagnostic algorithms to assign HPV causality in HNC easy to be implemented in the clinical practice, which is not the case for mRNA detection.

However, if this Reviewer still considers worthy to present these estimates in the manuscript after this justification we are willing to provide them.

    For the less specialist reader, the authors should explain clearly why they used E6*I mRNA, rather than E6 itself detection in their analysis.

An explanation on why we used E6*I mRNA, rather than E6 itself detection has been added to the “HPV E6*I mRNA analysis” section in material and methods as follows:

“An ultra-short amplicon, E6*I mRNA RT-PCR assay was chosen for HPV-mRNA detection for its applicability to FFPE material and absolute RNA specificity by using a splice-site as identification target [20].”

Reviewer 3 Report

This is a well designed and conducted analysis on HPV-driven HNSCC. The results are of major clinical interest und add to the literature published so far. There are no major changes needed to this manuscript. Well done.

Author Response

20-11-2020

Response to reviewers

Thank you very much for having reviewed our work. Please find below our answers and explanations on how we have addressed the Reviewers’ comments, point by point, in order to proceed with the submission of the revised version of the manuscript to Cancers. A new version of the manuscript, with track changes and clean, has been also prepared and added to the submission.

Reviewer no. 3

( ) I would not like to sign my review report

(x) I would like to sign my review report

English language and style

( ) Extensive editing of English language and style required

( ) Moderate English changes required

(x) English language and style are fine/minor spell check required

( ) I don't feel qualified to judge about the English language and style

          Yes    Can be improved     Must be improved    Not applicable

Does the introduction provide sufficient background and include all relevant references?

          (x)      ( )       ( )       ( )

Is the research design appropriate?

          (x)      ( )       ( )       ( )

Are the methods adequately described?

          (x)      ( )       ( )       ( )

Are the results clearly presented?

          (x)      ( )       ( )       ( )

Are the conclusions supported by the results?

          (x)      ( )       ( )       ( )

Comments and Suggestions for Authors

This is a well designed and conducted analysis on HPV-driven HNSCC. The results are of major clinical interest and add to the literature published so far. There are no major changes needed to this manuscript. Well done.

Thank you very much.